# Reflective Processes Promoted in the Practicum Tutoring and Pedagogical Knowledge Obtained by Teachers in Initial Training

Carolina Flores-Lueg 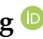

Department of Educational Sciences, Faculty of Education and Humanities, University of Bío-Bío, Avda. La Castilla 1180, Chillán 3820572, PC, Chile; cflores@ubiobio.cl

**Abstract:** This research focuses on the formative contributions made by tutor teachers in the university to the acquisition of pedagogical knowledge based on the reflective processes that they manage to implement, from the perspective of future teachers in professional practice or a practicum. A qualitative method was employed. Twenty practitioners of pedagogy programs who developed a practicum in the year 2020 or 2021 took part in this process, matriculated at three universities found in the Biobío and Ñuble regions in the central–south zones of Chile. A semi-structured interview was used as a technique and the analysis was carried out through categories of meaning with a phenomenological basis, whose topics were the following: (1) appreciation of the support of the tutor teacher, (2) strategies and resources used to promote reflection, (3) reflective actions linking theory and practice, and (4) types of knowledge obtained from the reflective processes implemented. Among the results, what stands out is that future teachers legitimize the guidance provided by the tutor and value the generation of reflective processes, but these are situated in an experiential and emotional dimension, stressing the formation of pedagogical knowledge. In conclusion, there is a need to redefine tutorial action and promote its professionalization.

**Keywords:** initial teacher training; practicum; tutorial action; pedagogical knowledge; teacher reflection

## 1. Introduction

For more than three decades in Latin America and the Caribbean, teacher training has become one of the central focuses of educational policies, because it is understood that the qualification of a teacher has a decisive impact on the learning results of students and, therefore, in the improvement of educational quality and the development of society [1–3]. Under this premise, governments have implemented a series of policies focused on promoting the improvement of the work of teachers who work in the school system, setting up mechanisms that emphasize their professional development from initial training, understanding this training level as the gateway to their professionalization [2].

In the case of Chile, initial teacher training (ITT) recently became the focus of educational policy, as it is only from the year 1990, with the return to democracy in the country, that state proposals focused on strengthening this aspect. Until now, all policies have been focused on the implementation of curricular adjustments or renovations in training programs; attracting the best applicants to teaching careers; the definition of pedagogical and disciplinary standards; the application of standardized evaluations to determine the levels of teaching performance; and the definition of regulations oriented towards the accreditation of universities and training programs [4,5].

Notwithstanding the above, ITT continues to be the object of multiple questions at the national and Latin American levels [1,4,6,7], proceeding from the argument that the preparation of teachers has not managed to produce the changes that are required to educate and learn in the 21st century [6,8], because the study programs continue to be mired in the supremacy of theoretical and disciplinary knowledge [9], the disarticulation

between theoretical and practical training [10], and the distance between academic and school realities [6,11,12]. Additionally, different approaches show a series of tensions in the training process generated by public policies, directed from economic interests that transcend the educational field [1,9,13], attributing to the teacher a technocratic position with a role of being didactic, for which the training focuses on the scientific–technical component of the trade [14].

It should be noted that, in the Chilean context, teacher training was transferred from normal schools to universities in 1974, the latter being the institutions responsible for training specialist teachers for the distinct levels of the school system (early childhood education, basic or primary education, and middle or secondary education), with study plans that last between four and five years. Although, in recent years, these have been renewed to respond to prevailing policies, overall, they present a certain homogenization, as they are structured in the following areas that group together a set of subjects: specialty or disciplinary training, pedagogical training, practical training, and comprehensive training [7].

A field that requires special attention is practical training, because that this activity is essential for learning how to teach, by offering the opportunity for students to start teaching in real and situated contexts [14], such that they can compare training in the profession with the reality of professional practice and integrate knowledge from theory with practice [15], as well as for the development of reflexive processes that allow them to obtain the necessary pedagogical knowledge for their future professional performance [16,17].

Within this scope, professional practice or a practicum constitutes the most relevant formative space and time, because it is expected that future teachers will be able to exercise their teaching role with propriety, receiving guidance from experienced professionals. One of them is a teacher assigned by a school, called a guide, mentor, or collaborator, and the other is a teacher assigned by a university, called a supervisor or tutor [8]. Even though there are research results on this subject that show that this training experience is highly valued by students in professional practice [7] and by the academic field [8,16,18], nowadays it is also the object of controversy because research shows that in its development, decontextualized and distant practices regarding the real requirements of school contexts prevail [6,19], approaching this activity from the perspective of technical application [8,20].

To move towards the improvement of ITT, the most current discourses argue about the need to assume training from a critical–reflexive approach, understanding that pedagogical action develops in a specific time and space in which political, social, and cultural factors converge, and that they can only be examined from immersion in the school reality. Under this perspective, a teacher is conceived of as a reflective professional with the ability to analyze and assess the diversity of situations, contexts, knowledge, and learning that they mobilize when developing their educational action [21], which allows them to generate professional learning and improve their practice [22,23]. Consequently, reflection becomes the key element that can transform the pedagogical practices implemented by teachers, and a greater reflective capacity that can be developed in initial training would be one of the guiding principles of quality training [24]. For this reason, it is necessary to seize the opportunities offered by the practicum and take them as an advantage in promoting the critical analysis of contexts by future teachers, activating the theoretical knowledge acquired in training as that which arises from their experience and collective teaching experience [25–27], to generate new professional learning that contributes to the improvement of educational practice [28].

Within the model of reflective practice, the role of the supervisor or tutor appointed is critical, because it is the experienced professional in the training triad who should guide and direct the student, ideally from a reflective and dialogic perspective, both personally and collectively [13,29]. This tutor assumes a privileged position by having a double vision of educational scenarios, as they know the dynamics and culture of the school context and, at the same time, they know the aims, demands, and administrative–pedagogical requirements set up by the university space. Therefore, it is assumed that they would be especially qualified to introduce reflective processes that allow the practitioner to

link theory and practice, putting into practice the theoretical schemes transmitted in the university [30–34]. Thus, a tutor's ability to promote critical reflective processes in future teachers would become an essential condition to favor the appropriation of a new repertoire of practical knowledge as a basis for the acquisition of pedagogical knowledge.

Although research on tutoring in the practicum is currently scarce and somewhat limited in the Chilean context [29], some studies have been found that focus on inquiring about the performance of this profession, evidencing the implementation of traditional tutorial practices [22] and the insufficient delivery of pedagogical support regarding the performances finally mobilized by the students in the practice contexts [19]. Additionally, recent research concludes that those who assume tutoring end up exercising their work from their professional experience and according to their criteria, because they do not have specific training in this field and their role is vaguely defined by the universities [35].

In addition to the above, research focused on the reflective processes of pedagogy students concludes that this action is presented in an instrumental form, mostly focused on the analysis of the design of linear classes in addition to the strengthening of weaknesses and breaks associated with the mastery of disciplinary contents [31]; additionally, it would be developed from a general point of view, situated from common sense and with a limited level of depth [30,36], without establishing relationships with the theory [37,38]. Finally, another study shows that both pedagogy students and tutors mobilize their actions from a technical rationality, which affects their reflection processes by only reaching low levels of questioning [23].

Taking into consideration that the basis of formative and pedagogical professional practice depends deeply on the tutorial action [38,39], this paper presents the results of research that had the purpose of exploring the meaning attributed by future teachers in professional practice to the contribution made by the tutor teacher in the acquisition of pedagogical knowledge from the reflective processes that they manage to implement. To respond to this, the following specific objectives were defined: (1) identify the value attributed by teachers in initial training to the actions implemented by the tutor teacher, linked to the promotion of reflective processes that contribute to the construction of pedagogical knowledge, and (2) characterize the pedagogical knowledge obtained by the future teachers, based on the reflective processes implemented by the tutor teacher.

## 2. Theoretical References

### 2.1. Teaching Knowledge, Pedagogical Knowledge and Reflection as Support for Teaching

The discourses on what teaching knowledge is and how it is manifested in pedagogical practice are extensive and diverse [40], as for several decades educational research has attempted to define a knowledge base for the exercise of the profession [41], showing in its generality that it responds to an epistemic framework made up of multiple types of knowledge that are articulated and mobilized in the teaching profession. Among the widely referenced theoretical proposals at the international level is the approach of Shulman [42,43], where the typification of the knowledge that is at the basis of teaching is essential to strengthen teaching performance, arguing that to teach a concept it is not enough to master the content and have general knowledge of pedagogy, but that specific knowledge regarding the teaching of that content is also required [44].

Shulman presents a model that combines two central elements. First is a knowledge base for teaching consisting of seven types of minimum knowledge: (1) knowledge of the subject matter taught; (2) general pedagogical knowledge; (3) knowledge of the curriculum; (4) pedagogical knowledge of the subject/content; (5) knowledge of learners and their characteristics; (6) knowledge of educational contexts; (7) knowledge of educational goals, aims, and values, and their philosophical and historical foundations [45]. Second is the identification of the main sources of acquisition of such knowledge: (1) academic training in the discipline being taught that enables students to communicate what is essential and what is peripheral; (2) didactic structures and materials that shape the principles underlying

teaching; (3) academic studies in education that relate to understanding the processes of schooling, teaching, and learning; and (4) knowledge acquired in practice itself [45].

From another perspective, Tardif points out that teaching knowledge is made up of five types of knowledge: (1) Professional knowledge, which is selected and transmitted through the curricula taught by teacher training institutions, and which is transformed into a type of knowledge intended for the scientific and scholarly training of teachers. The training institutions also try to ensure that this type of knowledge is incorporated into teaching practice. (2) Disciplinary, corresponding to the various fields of knowledge in the form of disciplines that arise from cultural tradition and social groups producing knowledge, which are transmitted in university courses and departments. (3) Curricular, referring to discourses, objectives, content, and methods on the basis of which the school institution categorizes and presents the social knowledge that it itself defines and selects as a model of scholarly culture. They are presented in the form of school programs (aims, contents, methods) that teachers must learn to apply. (4) Experiential or practical, those that teachers develop from the practice of their profession, which emerge from individual and collective experience in the form of habits and skills, of "knowing how to do and knowing how to be" [46]. They constitute knowledge that is validated in action and corresponds to a set of representations that serve as a basis for teachers to guide their profession and give meaning to their daily practice in all its dimensions. (5) Pedagogical: doctrines or conceptions stemming from reflections on educational practice, in the broad sense of the term, and rational and normative reflections that lead to more or less coherent systems of representation and guidance of educational activity [46].

When contrasting both approaches, it can be deduced that pedagogical knowledge of content is a category that can be homologated with pedagogical knowledge, as it constitutes the basis that gives meaning to the teaching developed by teachers, representing the mixture of knowledge about the discipline being taught and about how to teach it, and becoming a central category because it allows distinguishing between "the understanding of the specialist in an area of knowledge and the understanding of the pedagogue" [45].

In view of the above, it can be said that, in general, pedagogical knowledge can be understood as "knowledge in use" as it is made up of representations from which teachers orient their teaching action [41]. Therefore, it is linked to a knowledge of a practical, subjective, and dynamic nature, which is constructed, deconstructed, and reconstructed on the basis of the conjugation of cognitive, affective, and sensitive dimensions of the subject. It is a type of knowledge that is also acquired as a product of a local historical process developed from the daily interaction between teachers and students in a particular context [47]. Furthermore, although it is shown as a type of knowledge constructed outside the generalizable or abstract knowledge of science and positivist perspectives [45], it is the epistemic basis that gives meaning to pedagogical work.

Pedagogical knowledge reflects a type of practical knowledge of experiential origin and its acquisition requires reflective practice that allows teachers to analyze a particular teaching situation, moving from practice to theory and from theory to practice on an ongoing basis [25,28,44], to generate a combination of knowledge based on the link between both domains [25]. Therefore, reflective practice becomes a transcendental action for the acquisition of knowledge specific to the profession [28,46,48].

Given the relevance that reflective practice acquires for the construction of pedagogical knowledge, it becomes an action that needs to be promoted intentionally from initial training, through an accompaniment that allows the transformation of practical experience into professional knowledge [49,50], because a reflection mediated with a colleague or tutor "makes it easier to draw generalizations from practice, which can help to better identify the problems of practice, improve understanding and intervene more effectively" [51].

In addition to the above, it can be said that the main recommendations provided by recent studies highlight the need for teacher educators to ensure that future teachers acquire and mobilize the different knowledge they will need in real teaching situations [41]. Therefore, the role of the tutor is fundamental, as the characteristics of the pedagogical

support they provide to their students and the reflective practices they manage to promote are essential for this purpose

### 2.2. Approaches Underpinning Practice in Initial TeacherEducation and their Implications for Reflective Processes

Current models of practical teacher training support the idea of promoting learning to teach from a situational perspective, adapting pedagogical action to specific contexts and groups of students [24].

In the case of the Chilean reality, practice in initial teacher training is developed under the predominance of two epistemological approaches: a technical-applicationist approach and a critical-reflective approach [8]. Each of them has different implications for understanding the reflective processes that are implemented in practice scenarios.

From the technical-applicationist approach, practice is conceived as a training activity where the student teacher must apply and validate in the classroom teaching models designed by agents external to the school reality. In this perspective, theoretical and technical preparation precedes action, which would be sufficient for the knowledge acquired to be transferred to the teaching role [52]. Thus, the knowledge acquired in teaching is undervalued, which means that practical training does not acquire greater organization or structure on the part of the training institutions, because it is based on the premise that in order to learn to teach, it would only be sufficient to have good practice centers and good teachers to guide the trainee in the application of theory.

In contrast to the above, in the critical-reflective approach, teaching practice is seen as a complex and historically situated process, where the future teacher recognizes the complexities of the classroom and understands his/her pedagogical work from a reflective action that allows him/her to analyze his/her teaching with the aim of improving it, considering the variety of situations he/she faces in the classroom, together with the knowledge and know-how he/she mobilizes when developing an educational action. Under this approach, reflective practice is conceived as the mode of connection between thought and action, in order to develop new professional learning that allows teachers to improve their pedagogical work [48,52,53]. Thus, they become self-evaluators of their practice [53].

It should be noted that the notion of the reflective teacher is an old figure in the understanding of education [54], derived from Dewey's approaches, especially with the notion of reflective action. However, in the last 25 years it has become a focus of interest widely addressed in educational research [28,51], which has led this notion to be presented as a diffuse category, both in its meaning and in how to teach it [55]. In this regard, a study by Beauchamp [56] stands out, where, based on an analysis of conceptualizations associated with reflective practice present in the literature, it was determined that this concept can be referred to cognitive, metacognitive, or linguistic aspects; but it can also be linked to a problem-solving process or a process of social criticism [57].

One of the theorists widely referenced internationally on reflective practice is Donald Schön [48,57], who, taking into consideration Dewey's approaches, proposes the idea of the reflective professional, arguing that professional activity is not a model of applied sciences or instrumental technique, because, to a large extent, professional performance is improvised, constructed, and reconstructed during its development. Therefore, a professional cannot focus only on following "recipes" or applying theoretical knowledge derived from positivist perspectives to solve a problem, because each situation that is presented to him/her is uncertain and indeterminate, which requires him/her to reflect "in" and "on the action" to anticipate the solution of future problems and, at the same time, define them, redefine them, contextualize them, solve them, observe them from another perspective, and generate new knowledge [28].

The approach to the idea of the reflective professional, in its generality, has been situated in three traditions linked to the evolution of scientific rationality and teacher training [28]: (1) technical rationality; (2) reflective practice; and (3) critical reflection.

From technical rationality, teacher reflection is understood as a cognitive capacity that allows deliberation by making use of theory to provide solutions to practical pedagogical problems; from practical rationality, it responds to an exercise that generates a type of knowledge through experience in the light of theory (Schön, 2010), which makes it possible to generate new professional knowledge. Finally, critical reflection leads to the combination of reflective action with research to transform practice, considering the political and social factors that condition the work of teachers [58].

### 2.3. The Tutorial Action and Its Relevance in the Practicum

The pedagogical accompaniment implemented by the tutor teacher is valued as one of the essential components for training [59–62], because the tutor teacher becomes the facilitator of the learning processes of the trainees so that they can progressively perform in a more independent manner over time, understanding that it is not enough to control or supervise but that contributing with perspectives of analysis and knowledge that should normally enrich the student's practical experience and stimulate their professional development is required [15]. To do this, the tutor teacher must have a broad proficiency in the pedagogical process that takes place in the school where the teacher in training will be inserted for the development of their practices [59]; however, in addition, they must meet a series of professional functions, including diagnosing, guiding, controlling, and mediating the training process in a continuous and systematic way [60].

In relation to the roles that the tutor teacher must assume, in the most current literature the following five roles are identified: (1) accompany the intern in their passage through the school; (2) evaluate them so that they achieve the objectives and qualify them; (3) in a socioaffective sense, provide support in the face of emotional situations associated with the practice; (4) act in a pedagogical/didactic/disciplinary manner in terms of the interactions between the tutor and the student, facilitating the mobilization of professional skills in context according to the discipline of the curriculum; and (5) make links between theory and practice, closely related to reflective and professional development [61]. The first three are the most common, while the other two are found the least often.

For the above, it is considered important to highlight that for the tutorial action to respond to a training process adjusted to the current performances demanded from the teaching staff, it is required for the tutor to have exhaustive preparation in teacher training, and this implies being in a process of constant updating in the different dimensions of pedagogy and educational policies, along with having personal qualities that allow them to fulfil their role, including empathy, flexibility, and communication skills. Specifically, it is necessary to advance towards the professionalization of it, re-signifying its formative function and its pedagogical practices [60].

## 3. Materials and Methods

### 3.1. Research Focus

This study was carried out with a qualitative methodology under an approach of hermeneutic phenomenology [63], because the main interest was focused on understanding the essential meaning of phenomena as they are presented in experience and in the world of life to the subjects, as well as the sense and meaning that they have in the purpose of the context of application for their interpretation [64]. It was of special interest to investigate the meaning attributed by pedagogy students in professional practice to their experiences related to the actions of the tutor teacher, specifically with regard to those interactions in which reflective processes that contributed to the construction of pedagogical knowledge were promoted.

### 3.2. Study Context

For the development of the research, three universities belonging to the Consejo de Rectores de las Universidades Chilenas (CRUCH) were considered, located in the Biobío and Ñuble regions in the central–south zones of Chile, which teach the programs

of Pedagogy in Early Childhood Education, Pedagogy in Basic Education, Pedagogy in English, and Pedagogy in History and Geography. These institutions were selected based on the determination of convenience criteria [65], considering accessibility and proximity of access to the field of study.

For accessibility, a formal letter was sent to the deanships of each faculty of education, in which the purposes of the study and aspects related to the method were explained, in addition to requesting the corresponding authorization to enter the field of study. Once the answers were obtained, contact was made with the administration office of each pedagogy program to explain the scope of the research and to request their collaboration in contacting possible participants who met the following criteria: (1) be a student of the programs of Pedagogy in Early Childhood Education, Pedagogy in Basic Education, Pedagogy in English, and/or Pedagogy in History and Geography; (2) be developing professional practice during the year 2021 or have carried out this activity in the year 2020; and (3) agree to participate voluntarily and disinterestedly in the study, expressed by signing an informed consent. From the information provided by the administration of these careers, a database with the names and emails of the potential participants was generated, to which only the researcher has access.

### 3.3. Participants

Based on the above criteria, twenty teachers in initial training for pedagogy programs belonging to the selected universities took part. Their distribution is shown in Table 1.

**Table 1.** Distribution of participants by university and pedagogy programs.

| University | Pedagogy Programs | | | | Total (Institutional) |
|---|---|---|---|---|---|
| | Early Childhood Education (ECE) | Primary Education (PED) | History and Geography (PHG) | English (PE) | |
| University 1 (region of Ñuble) | 8 | 2 | 3 | 1 | 14 |
| University 2 (region of Biobío) | 4 | 0 | 1 | 0 | 5 |
| University 3 (region of Biobío) | 0 | 1 | 0 | 0 | 1 |
| Total | 12 | 3 | 4 | 1 | 20 |

Source: Prepared by the author.

In response to the situation of social distancing generated by the COVID-19 pandemic, each individual was initially contacted through an email in which the purpose of the study was explained, and they were asked to voluntarily supply their telephone numbers to discuss the extent of their participation and, at the same time, establish a date and time to schedule an interview. Those who agreed to participate received a new email requesting them to sign the informed consent.

As can be observed in Table 1, significant differences can be seen in the number of participants between the three universities considered. This happened due to the total number of informed consents that were returned with the respective signatures. It should be noted that 85% of the participants were women and 15% were men; 20% developed their professional practice in 2020, and 80% were developing this activity during 2021. The age range was between 22 and 26 years old.

### 3.4. Techniques Used

To obtain information, a semi-structured interview was used, understood as a technique that allows the obtaining of descriptions of the world of people's lives with respect to the interpretation of the meaning of the phenomena described [66]. Prior to its application, a script of questions was prepared that had the topics within the study defined based on the aims, and initial questions were established to guide the interview (see Table 2).

**Table 2.** Topics considered for the interview and script of questions.

| Specific Objectives | Topics | Questions |
| --- | --- | --- |
| Identify the value attributed by teachers in professional practice to the actions implemented by the tutor teacher, related to the promotion of reflexive processes that contribute to the construction of pedagogical knowledge. | - Appreciation about the support of the tutor teacher.<br>- Strategies and resources used to promote reflection. | - Can you tell me about the support provided by the teacher tutor?<br>- In what moments do you feel that the tutor leads you to reflect on the practice?<br>- What does the tutor do to make you reflect? |
| Characterize the pedagogical knowledge obtained by the future teachers, based on the reflective processes implemented by the tutor teacher. | - Reflexive actions that link theory and practice.<br>- Types of knowledge obtained from the reflective processes implemented. | - At what moment have you been able to analyze your pedagogical work considering some theoretical knowledge obtained in your training?<br>- How does the tutor help you connect your practice with theory?<br>- What situations experienced with the tutor have allowed you to obtain new knowledge for your teaching action? |

Source: Prepared by the author.

It is worth mentioning that this technique was applied to each individual directly by the researcher. The Zoom platform was used, given the situation of social distancing. The day and time for the interview were previously arranged by telephone with each participant, taking into full consideration their time availability and Internet access possibilities. At the time of the virtual meeting, prior to the interview, a brief informal conversation took place, with the purpose of generating a more relaxed and fluid environment. Once the interview began, the purpose of the study was explained again, and guidance on how it would develop was provided. Each participant was asked to have their camera turned on if possible and for authorization to record the interview. Each meeting lasted approximately 50 min, and the interviews were transcribed verbatim.

*3.5. Data Analysis*

An inductive procedure based on phenomenologically based meaning condensation was used, considering the following five steps: [66]

(1) Full reading of each interview: Once each of the interviews had been transcribed, they were read individually to explore the content of the narratives produced by the participants and to have an idea of the whole. In this first step, we also took advantage of contrasting the literacy of the written document with the recordings of the interviews to confirm that there were no errors in the transcript.

(2) Determination of units of natural meaning of the texts, as expressed by the subjects: Each interview was read a second time to decide the units of meaning as expressed by the participants. Certain extracts of texts (quotes) were demarcated, attaching memos to them.

(3) Formulation of the theme that dominates a unit of natural meaning: the content of the demarcated quotes was examined and then the dominant theme within the unit of meaning was defined, operationally assigning it a code.

(4) Interrogation of the units of analysis based on the purpose of the study: the themes defined in the units of meaning were analyzed from the point of view of the purposes of

the study, reflecting on their relevance. Each of the interviews was contrasted with the purpose of looking for convergences between the meanings of the participants.

(5) Link nonredundant topics in a descriptive statement: in this step, the selected topics were analyzed again and relationships between them were sought in order to link them in a descriptive statement (see Figure 1).

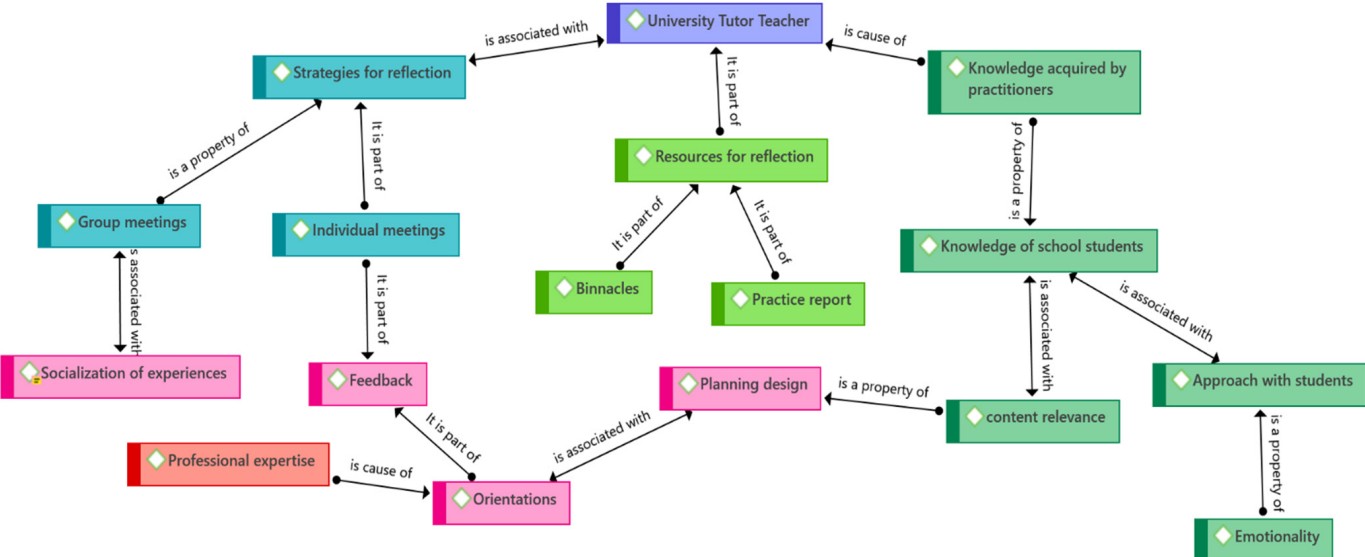

**Figure 1.** Thematic network: promotion of reflective processes by the tutor teacher and the knowledge obtained in the practicum. Source: prepared by the author based on data supplied by Atlas. Ti. v. 9.

The entire analytical process was supported by Atlas.ti v. 9 software and to safeguard the quality of the data analyzed, intersubject triangulation was used as a complementary procedure, looking for convergences and divergences of meanings expressed in the discourses [67]. Additionally, an attempt to "put in parentheses" (Epojé) the theories of the researcher was made throughout the application process of the interviews [66], to achieve a natural understanding of the experiences lived by the teachers in training in the development of their professional practice and the meanings granted.

### 3.6. Research Ethics Codes

The participation of the individuals was protected according to the following ethical codes:

(1) Signature of informed consent prior to agreeing to take part in the interview. This document was approved by the Ethics and Biosafety Committee of the institution in which the researcher works under a Certification No. 011/19 certificate that was accepted by the funding agency of the research project, the Agencia Nacional de Investigación y Desarrollo de Chile (ANID).
(2) Request for authorization prior to recording the interview.
(3) Anonymity in the transcript: protection of the identity and intimacy of each participant and of the university considered as the study context.
(4) Transcript: the transcription of the interviews was conducted verbatim, and an attempt was always made to make it as faithful as possible to the original.

### 4. Results

The results obtained regarding the reflexive processes implemented in the tutorial action are presented below, characterizing the strategies and resources used by the tutor teacher from the perspective of the teacher in training, as well as the knowledge that they

say they were able to obtain in the development of this exercise. For this, the codes that appeared from the steps used in the data analysis were considered (see Figure 1).

*4.1. Value Attributed by Future Teachers to the Actions Implemented by the Tutor Teacher in the Promotion of Reflective Processes*

4.1.1. Collective and Individual Meetings as a General Strategy That Promotes Reflective Processes

Teachers in training widely emphasize that collective and individual meetings formed general strategies that allowed them to reflect on their pedagogical performance.

Collective meetings were held once a week, and there is evidence of convergence in the speeches of the participants by pointing out that their tutor began the meeting by providing technical–administrative information and general guidance on practice, but, after that, offered time to share experiences or situations experienced in the school. That moment of weekly socialization is extensively valued as the instance in which they could reflect, because the opportunity to listen to the experiences of their peers helped them to examine their own experience and performance. Additionally, it helped them to contrast the different realities of practice and extract some ideas about strategies and/or resources that provided results for others. At the same time, these meetings became a space where they could freely express their feelings about the process, a formal instance to "let off steam" and receive guidance on how to improve. Additionally, although the speeches highlight that these meetings were quite reflective, the internal dynamics and the situations narrated by the participants make it seem as though the reflective action was situated within an experiential and emotional scope, because it involved relaxed conversations regarding what they experienced in the school, as shown in the following quotes:

> It is good to compare the processes of my classmates, sometimes to see how they are all different realities, and sometimes we do not tend to talk about our feelings, our emotions respect to the process, but it is very gratifying to be able to talk about this with the teacher and to receive guidance to improve. [21:11 - 39 in Interview 2/ECE]

> It was still good to let off steam, but I don't know if it was so theoretical, everything was much more relaxed, it was much more spontaneous, it happened. [20:16 - 43 in Interview 11/PED]

The individual meetings were personalized encounters between the tutor and the intern, and they were held right after direct observation in the classroom once or twice a month. In this interaction space, the tutor provided feedback on what was observed in class, presenting comments on what they liked about the performance and what should improve, while also supplying guidance on how to perform it better. This feedback instance allowed the interns to reflect on their pedagogical action and to become aware of certain errors made in their performance that were not self-perceived at the time. At the same time, they value positively the indications on the design of the planning and the delivery of certain suggestions to manage their classes in the school, as shown in the following examples of quotes:

> The teacher gave us some advice on how we could handle such and such a situation in case it arose, and she gave us other advice about we planned, and as we were just learning this, we were still missing some points and she was reinforcing them. [15:31 - 25 in Interview 7/PE]

> All time I was being evaluated and the truth is that it is something that I really appreciate because when someone is teaching there are many things that one does not realize, that may not be correct or that I have not explained well or even used the right words, so, the role of the teacher has been very important. [27:12 - 45 in Interview 9/ECE]

> When she went to observe classes or came in, always told me . . . "you are missing this, I liked your class, but you lacked such a thing, you should do it like this",

or she gave me options, so well, they have been a great support. [29:10 - 15 in Interview 13/PED]

Therefore, it can be deduced that these instances of individual and collective meetings were viewed by the teachers in training as the spaces in which they could reflect on their practice experience and on their classroom action, based on two specific actions implemented by the tutor: the feedback of the performance observed and the generation of spaces to socialize experiences of practice.

4.1.2. Resources Used by the Tutor Teacher to Promote Reflection

According to the future teachers, the tutor used some resources that allowed them to reflect among themselves; the registration of daily or weekly logs was widely highlighted, because it forced them to stop and think about what they were doing in their classes, depicting their perceptions based on certain criteria. This is how they pointed out the following:

We also had to create weekly reflection logs which meant that you had to take any of the experiences and reflect based on different criteria. [20:9 - 39 in Interview 11/ECE]

We had to deliver pedagogical reflection logs, so we analyzed experiences that we had during the week and that was very good because sometimes there is so much to do on a day-to-day basis, that there is no time to stop and analyze what you are doing, so in those blogs I think it is where I reflected the most. [23:11 - 23 in Interview 4/ECE]

In addition, some of the participants also referred to the preparation of a final practice report that was graded, but as can be seen in the following reports, its nature was descriptive:

You have to hand in a reflection report on the practice, you have to give some data of the establishment and specific data of the course in which we developed our practice and put, for example, our appreciation of how we felt the course took the English classes, how the course perceived them, what we thought, it was that. [15:55 - 105 in Interview 2/PE]

They also asked us to make a report, it was more general, it was more contextualized to the establishment, like our pedagogical work, but from a very superficial perspective. [20:10 - 41 in Interview 11/ECE]

At the end we had to make a full report of our practice and that's where the qualification came in. [2:98 - 33 in Interview 1/PHG]

*4.2. Knowledge Obtained/Reinforced in the Development of the Practicum with the Support of the Tutor Teacher*

4.2.1. Theoretical Knowledge and Tutorial Action

In particular, in the speeches, the theoretical knowledge obtained in the study of psychology was emphasized. From the perspective of the practitioners, these have been the most useful within the practice. Specifically, they allude to the characteristics presented by the stages of human development, the knowledge of strategies to capture the attention of students, and certain "tips" on how to interact with them, which can be seen in the following quotes:

Having these theoretical bases is very useful, also the subject of psychology in children, in adolescents, is a subject that is very useful when teaching, because one can already think about what one can do to capture the attention, or to achieve that there is learning on the part of the students. [15:47 - 87 in Interview 7/PE]

Now one is in charge of the course, and there I tried to remember what the educational tips of educational psychology were, how I had to show myself to the student, how I could react. [17:30 - 75 in Interview 9/PHG]

In addition to the above, the stories of two participants stood out, who explicitly referred to the application of constructivist theories of learning in their pedagogical action. One of them points to Vygotsky and the other to Ausubel, arguing that their postulates have served as a framework within their teaching practice. As can be seen, the reflective process implemented has allowed them to become aware of some actions that could improve or corroborate their scientific validity in practice. Some examples of narratives are the following:

> I remembered what Vygotsky said about the zone of proximal development, about having someone who helps you obtain knowledge but does not give it to you, as simple as that. I related that as I was developing learning experiences [ . . . ], sometimes the principle fell into giving the answer very easily if someone did not respond and giving more signals, and then I analyzed it, and said . . . I must stop a bit; I must ask better questions. [23:17 - 37 in Interview 14/ECE]

> I used the island of Juan Fernández when I told them about the introduction of foreign species on the island and the damage it caused, and if you only knew! . . . you cannot imagine many conversations, conclusions, reflections, and everything that my students got! And one of the things that I highlight the most is that at one point in a class, a student who did not participate much told me . . . "You know what, teacher? I always see everything you have said in class in Curanilahue," which is where he was from, "I see how people building their houses, paving and everything else have diverted what used to be the river", he told me, "you look at it and it's like everything that the same people led it as they wanted, I look outside of the city and I only see pine trees teacher". It also happens to me where I am from, I am from Cabrero, so I look around and there are pine trees everywhere, and he said "teacher, everything you say happens to me" . . . when the student told me that and when other students told me about what they lived, I understood what Ausubel was saying. It was all associated with their own mental processes, with their earlier experiences, with what they saw and lived with what they knew. It is the closest thing I had to the implementation of significant learning. [16:115 - 56 in Interview 8/PHG]

It should be noted that only these two participants supplied these types of stories, and, when discussing whether they had been intended by the tutor teacher, only one of them commented on the following:

> My supervising professor helped me in this area, in the construction of this type of reflection. His help was especially useful. I take this into consideration for my future work. [16:116 - 40 in Interview 8/ PHG]

In the speeches of the rest of the participants it was not possible to identify reflections in which the teaching practices were examined from some theoretical model, since in the reports it is evident that during the interactions with the tutor the dialogues generated and the support provided came mostly from comments on the experiences of the practice, the analysis of technical documentation (planning), and the delivery of certain types of theoretical material linked to some situation presented in the classroom, in the face of which they felt that they did not have enough preparation. Here are some examples of their stories when asked about how the tutor teacher helped them link their practice with theory:

> More than anything, he guided us, for example, in the bases, he went over the issue of planning, from where to get certain resources. [24:37 - 53 in Interview-ta15/PBE]

> My teacher reminded us of every week, she gave us workshops too, we reviewed the plans, she asked us how our interaction was, besides, she was also going, I mean, she went to a class to observe, so after that she could give us feedback, she explained to us what was good, what we could improve. [23:7 - 9 in Interview 14/ECE]

> I truly value that my tutor focused on how we plan, how we deliver the content, if there were indicators that she considered were not right. [27:11 - 41 in Interview 18/PEP]

> The other day a classmate said she had problems because a boy at her school, from her class, could not control his anger. He had very little tolerance to frustration, things like that, and at that moment, she told us "Hey! Look, I have this" and she showed us some brochures from Chile Crece Contigo, where it said how to manage these situations. Later on, she sent us the material so that we could read it. [26:21 - 46 in Interview 17/ECE]

4.2.2. Practical Knowledge Obtained with the Support of the Tutor

According to what was said by the participants, the knowledge obtained in the practicum with the support of the tutor is fundamentally of a practical nature, that is to say, from their professional experience within the reality of the school environment, and it is associated with the contextualization of the contents in the classroom reality. They recognize the relevance of the knowledge of the course and the interaction they manage to generate with their students, beyond the planning and approach of the discipline. In the same way, they point out that important learning achieved with the support of the tutor is related to the relevance of the emotional dimension involved in the student's learning process. This knowledge, obtained from the reflective processes generated in the interactions with the tutor teacher, would become the basis for the exercise of their future teaching work, as can be observed in the following quotes:

> The teacher is a specialist in the orientation area, so more than anything else the topic was almost always focused on the emotional area, the children's area of self-knowledge, how to work on that side [ . . . ], in fact, I work a lot on the emotional development of my little ones and with that we are working on what is the specialty subject. [28:16 - 61 in Interview 19/PED]

> The most significant thing that I had to learn was to organize all that full time as a teacher, it is what I could get the most meaning out of my practical experience, because it is what made me understand the most, understand and retrieve from my performance as a teacher. [16:103 - 16 in Interview 8/PHG]

**5. Discussion**

The participants valued the support they received from the tutor teacher, essentially the suggestions provided regarding the design of plans relevant to the group of students and the recommendations for improving classroom performance. From the results of the analysis, the recognition of the tutor teacher's professional experience, the knowledge they have about the school context, and the practical knowledge that they transfer to the future teachers, which is assumed, without any type of questioning, to apply in the classroom, can be observed. Thus, the tutor becomes a legitimate pedagogical authority for future teachers, based on their own accumulated experience in the exercise of the profession, which contributes to the acceptance of their orientations by practitioners, renouncing other possibilities of action [68].

Regarding the strategies and resources used to promote reflective processes, collective and individual meetings are the fundamental times and spaces for this action. Within the collective meetings, the spaces granted by the tutor to socialize experiences of the school center are widely highlighted, because by listening to the experiences of their peers tutors can reflect on their own performance [7], and obtain ideas about strategies and/or resources that can be used to support their classes, but also share feelings associated with the development of their practice within a relaxed space for conversation. Although these socialization spaces helped them reflect on their teaching, in the results it is possible to observe, in general, that the reflective processes that they managed to develop were

fundamentally situated within the experiential and emotional dimension [69], lacking instances that would allow them to link theory with practice and vice versa.

On the other hand, it is evident that the feedback provided by the tutor in the individual meetings is transformed into actions that allow practitioners to reflect on the performance, because it helps them to become aware of the errors made and, at the same time, they can receive instructions on how to amend them. According to the results, the comments provided by the tutor moved between affective and cognitive aspects, recommendations on what the practitioner should do to achieve a more effective didactic action predominating within the latter [70,71]. This result coincides with other studies that conclude that, in the practice scenario, the emphasis on learning to teach from doing, leaving reflection in the background [46], complicates the possibility of setting up the link between theory and practice as well as between the university and the school [71].

Among the resources used by the tutor teacher to develop reflective processes, a positive assessment is observed towards the registration of logs, since the participants point out that they had to focus on describing certain precedents based on previously defined criteria, but, in addition, they recognize that they should think critically about their own performance and the relevance of their pedagogical practice. Although the elaboration of a practice report is also highlighted, the question remains as to whether the meaning of this document was focused on developing reflective processes on the results obtained in its pedagogical intervention within the school, or if it only constituted a document required to comply with one more grade within the subject, since the information demanded by the practitioner was generally informative–descriptive, with a self-assessment of performance with an emotional component.

The reflective processes generated, both from the tutor teacher and from the tutors themselves, mostly move from a technical level to a practical one. From the technical dimension, the reflection is centered on the design of the plans and the selection of those resources that allow better results to be achieved, and, from the practical dimension, the reflection advances towards the analysis of the experience and the contextualization of its pedagogical action through the needs of the students of the school, but it lacks a critical analysis in which reflective action articulates the experience, the pedagogical practice, and the contextual situations from a theoretical position or vice versa [36–38]. It is interesting that only two participants managed to advance towards deeper levels of reflection by placing themselves within a theoretical framework of reference for the analysis of their pedagogical action, one of them by personal initiative and the other with the support of the tutor teacher.

Although some tutors provided certain theoretical support to the students to deal with emerging situations in the classroom, this action was of a technical–instrumental nature, because it occurred occasionally and only to fill certain knowledge gaps manifested by the trainees. Even the responsibility of considering the material supplied and using it as a theoretical background to address future situations was completely delegated to the practitioners, since the generation of reflective spaces that would allow them to analyze the theoretical support supplied is not observed in classroom situations. Consequently, the link between theory and practice/practice and theory constituted an action that was finally conducted via personal initiative, rather than an action intended by the tutor. Therefore, future teachers would be the ones facing the challenge of articulating theory with practice within the reality of the classroom [70], and, with this, there is a risk that they will not be able to stress their representations, beliefs, and certainties about teaching installed since before entering the pedagogy program [61].

Regarding the knowledge obtained by future teachers in the practicum that related to the design of planning, the use of certain resources for teaching and the characteristics of children and young people stands out. In part, this knowledge derives from the transfer of the practical knowledge that the tutor has, which is mostly mobilized in the moments of individual feedback, from the instances of the socialization of practical experiences, and from the personal reflective processes developed during the experiences in the school.

Therefore, it can be said that, in general, this is knowledge constructed and reconstructed from the experiential field, which, from a practical dimension, is barely managed to be articulated with theory.

Finally, if it is taken into account that teaching reflection requires a questioning process that must be accompanied by concepts and theories to incorporate new points of view and support new ways of implementing pedagogical action [70,71], the reflections developed by the tutor from experience and practical knowledge stress real possibilities for the tutored to be able to advance towards the construction of pedagogical knowledge, that is, that type of plural knowledge that is re/configured from the convergence of the theoretical, experiential, personal, and contextual dimensions, but is based on reflective processes that allow dialogue with a particular teaching situation to give rise to the emergence of new professional knowledge [28,59]. If it is not possible to promote more complex and profound reflection in this formative instance, there is a risk of turning teaching into a meaningless task, into activism without substance [70], leaving the tutorial action and the pedagogical action of the tutor submerged within a technical level [23].

## 6. Conclusions

With regard to the first specific objective, future teachers value positively the actions implemented by tutor teachers in terms of support, suggestions, and the promotion of reflective processes generated in the development of the practicum. However, this last action would only be contributing to the construction of practical knowledge sustained from strong experiential and emotional aspects, since the reflective actions implemented are characterized by a fundamental focus on the analysis of practical experiences and the emotions that emerge from them. This level of reflection leads to perpetuating the gap between the theoretical training received at the university and the practical knowledge obtained at the school, as well as reducing the chances that future teachers will be able to build an epistemic base that acts as support for their future pedagogical decisions autonomously.

In addition to the above, in order to move towards a model of critical–reflexive practical training that contributes to the construction of pedagogical knowledge in future teachers, it is necessary to have prepared tutor teachers, capable of displaying not only their personal and professional abilities to generate the support required by the practitioners, but who must also be trained to develop reflective processes that allow the engagement of the practical knowledge that the trainees are acquiring in the school space with the theoretical knowledge transmitted by the university, in order that it can contribute to the generation of a repertoire of professional knowledge that will serve as a basis for teaching, based on pedagogical knowledge that gives meaning to their teaching performance. Specifically, universities must enhance the figure of the tutor teacher and assume the challenge of supporting their work by offering instances of professional preparation for the exercise of their function, because this action cannot only be situated from personal criteria and from the tutor's own representations, but requires specific training and monitoring of the role.

Finally, continuing investigation of the tutorial actions that are implemented in the practicum is recommended, with an emphasis on the sense and meaning attributed to the reflective processes from the perspective of the university professors who assume that function. This research can help to understand the way in which these actors understand and value their professional role, as well as exploring the way in which they think about educational activity. This would allow for systematized knowledge to formulate lines of action and/or support based on empirical data that serve as a base to contribute to the improvement of their function and initial teacher training in a curricular activity as relevant as it is to professional practice.

**Funding:** This research was financed by Fondecyt Project N°11190477, ANID-Chile: Professional practice and its contribution to the construction of pedagogical knowledge in future teachers: a study from the formative triad, and by the DIUBB GI/VC Project N°195723 PROFOP Research Group: Teaching Staff and Training Policies, Universidad del Bío-Bío, Chile.

**Institutional Review Board Statement:** The study was conducted in line with the Declaration of Helsinki and approved by the COMITÉ DE BIOÉTICA Y SEGURIDAD UNIVERSIDAD DEL BÍO-BÍO (protocol code 011/19, dated 10/10/2019) for studies with human beings.

**Data Availability Statement:** Not applicable.

**Conflicts of Interest:** The author declares no conflict of interest.

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
