# Peer review of "Reflective Processes Promoted in the Practicum Tutoring and Pedagogical Knowledge Obtained by Teachers in Initial Training"

_education, doi:10.3390/educsci12090583_

Round 1
Reviewer 1 Report
The authors take and explore a relevant and important topic that concerns development of teacher knowledge during practicum. I find the study interesting especially given that teacher knowledge comprises knowledge components and skills that are largely acquired in and with practice. That said, I also have noted some aspects that must be worked on to improve the quality of the paper.
1. An extensive editing of the English language is required. This problem clearly makes it difficult to study and understand what is being communicated throughout the entire report.
2. The materials and methods part needs to be improved further by providing more information about contents of the "in-depth interviews" as the author claims. That's the contextual information regarding what kind of questions were used to capture the meanings emerging from the reflective process of teachers (i assume these are teacher trainees). Moreover, it is not clear what this reflective process is about or when it is triggered or how it is triggered prior to the interviews. one is left wondering about the contextual details that are left out.
3. Since there is no detailed information about the nature of collected data or what kind of information or responses gathered with the interviews, the analysis part provided is very brief showing the steps from Kvale 2011. Moreover, the author claims that he/she employs hermeneutic phenomenology as a method. I find this problematic since the author states this as simply a method. I wonder if the author implies it as a method for analysis or for something else? The reader would benefit from getting more about what constitutes this hermeneutic phenomenology and how it is realized in this study
4. since there is limited information regarding the nature of data collected, it is difficult to understand why the author comes up only with two subheadings as the results/findings from the analysis. It is somehow difficult for me to tell whether or not the presented findings are all that could come from the data. Perhaps having well stated research questions or a clearly stated purpose of the study would solve this problem.
Author Response
Point 1: An extensive editing of the English language is required. This problem clearly makes it difficult to study and understand what is being communicated throughout the entire report.
Response 1: It is sent to a new translation using the service offered by the journal
Point 2: The materials and methods part needs to be improved further by providing more information about contents of the "in-depth interviews" as the author claims. That's the contextual information regarding what kind of questions were used to capture the meanings emerging from the reflective process of teachers (i assume these are teacher trainees). Moreover, it is not clear what this reflective process is about or when it is triggered or how it is triggered prior to the interviews. one is left wondering about the contextual details that are left out.
Response 2: The relevance of the in-depth interview as a technique is reflected upon and the conclusion is reached that it was rather a semi-structured interview when considering previous questions to guide the interview. The modification is made and the questions used are incorporated into the article. Indeed, it is about teachers in training, it is the concept that is currently being used in the Chilean context to refer to students of pedagogy careers.
An attempt was made to clarify from where the reflective process is being understood by incorporating more background into the problem and adding a section with theoretical references.
Point 3. Since there is no detailed information about the nature of collected data or what kind of information or responses gathered with the interviews, the analysis part provided is very brief showing the steps from Kvale 2011. Moreover, the author claims that he/she employs hermeneutic phenomenology as a method. I find this problematic since the author states this as simply a method. I wonder if the author implies it as a method for analysis or for something else? The reader would benefit from getting more about what constitutes this hermeneutic phenomenology and how it is realized in this study
Response 3. Being a qualitative study, the nature of the data collected responds to the construction of senses and meanings attributed by the participants to the phenomenon under investigation. The questions formulated are open and referential, which also allowed to investigate beyond the answers provided by the participants, from the formulation of new questions. The way in which the steps proposed by Kvale (2011) for the analytical process were used was detailed. You are right about the use of hermeneutic phenomenology, rather it was an approximation to the methodological proposal of van Manen (2016), understood from the phenomenology of teaching practice within the world of life in which it develops. More background information is incorporated into the methodology.
Point 4. since there is limited information regarding the nature of data collected, it is difficult to understand why the author comes up only with two subheadings as the results/findings from the analysis. It is somehow difficult for me to tell whether or not the presented findings are all that could come from the data. Perhaps having well stated research questions or a clearly stated purpose of the study would solve this problem.
Response 4. The corresponding adjustments were made in the presentation of the results, better organizing the subtitles taking into account the themes, the questions used in the interviews and the figure of the network generated from the analytical process.
Reviewer 2 Report
It was interesting to look into how future teachers (teachers in training) think about their tutor teachers with regard to the tutorials to promote reflective processes. Also, the data collection and analysis methods seem appropriate to meet your study purpose. However, there is much room for improvement:
- More description or information about the research design, data collection processes, and analysis methods would have been helpful to validate your findings and argument. For example, details of reflective processes implemented in tutorials, more information about the in-depth interviews (interview questions used, interview structure, who conducted the interview, how they were trained for the interview, what kind of efforts were made to minimize possible bias from the interview data, etc.)
- You might want to include more data and a thick description of it in your results section to support your argument in the discussion and conclusion section.
- It might have been helpful if you could add references to other similar studies that looked into student teachers’ responses or reactions to their tutor teachers in various contexts to see how your study finding is aligned with or contradicting those previous studies (either in the Literature review section or Results/Discussion section)
- There were too many run-on sentences and sentences with grammatical errors. You are encouraged to include one idea in one sentence to make your point clearer.
- It is not explained what the symbols next to the arrows in Figure 1 are indicating.
- There's an error in the number in Table 1.
Author Response
Point 1: More description or information about the research design, data collection processes, and analysis methods would have been helpful to validate your findings and argument. For example, details of reflective processes implemented in tutorials, more information about the in-depth interviews (interview questions used, interview structure, who conducted the interview, how they were trained for the interview, what kind of efforts were made to minimize possible bias from the interview data, etc.)
Response 1: The research design used is better detailed, attending to each of its recommendations.
Point 2:. You might want to include more data and a thick description of it in your results section to support your argument in the discussion and conclusion section.
Response 2: More information on the research problem is incorporated, a section with theoretical references is added, the results presented are organized in a better way.
Point 3. It might have been helpful if you could add references to other similar studies that looked into student teachers’ responses or reactions to their tutor teachers in various contexts to see how your study finding is aligned with or contradicting those previous studies (either in the Literature review section or Results/Discussion section)
Response 3. Observation is attended to and more background on previous research on the problem is incorporated and an attempt is made to contrast the results of the study with these in the discussion.
Point 4. There were too many run-on sentences and sentences with grammatical errors. You are encouraged to include one idea in one sentence to make your point clearer.
Response 4. Grammatical adjustments are made to the original article and it is sent to a new translation using the service offered by the journal
Point 5. It is not explained what the symbols next to the arrows in Figure 1 are indicating.
Response 5. The requested adjustments are made.
Point 6. There's an error in the number in Table 1.
Response 6. Correction is made
Reviewer 3 Report
While this is a worthy contribution to the field of education, this article requires extensive editing of English Language usage. The lack of clarity impeded the comprehension and understanding of this research.
I would suggest that the authors submit the original Spanish language article to a professional translator, then resubmit. In its present form, this article is very hard to understand.

Author Response
Response to Reviewer 3 Comments
Point 1: While this is a worthy contribution to the field of education, this article requires extensive editing of English Language usage. The lack of clarity impeded the comprehension and understanding of this research.
Response 1: I am very sorry that the translation of the article was not of good quality.
Point 2: I would suggest that the authors submit the original Spanish language article to a professional translator, then resubmit. In its present form, this article is very hard to understand.
Response 2: Suggestion is accepted and the article is sent to a new translation using the service offered by the journal
The observations made within the attached document are taken into account and an attempt is made to correct each of them.
Reviewer 4 Report
The term 'reflexive' needs to be changed in the title to 'reflective' - these two words are not interchangeable and do not mean one and the same thing
Review keywords beneath the abstract
Strengthen theorisation underpinning the use of such concepts as reflection, reflective practice, professional knowledge, pedagogical knowledge and key tenets of tutoring
Strengthen methodology section - hermeneutic phenomenology requires greater clarity and depth; what does the literature say about in-depth interviews; what are the ethical issues related to the use of zoom as your platform and how were these mitigated; how was data stored and confidentiality, anonymity assured; what questions were the participants asked ...
Section 3 - how were the narrative excerpts selected (beyond the use of computer software); which pedagogical programme and university did your interview participants attend ... with approximately 50 minutes (x20 participants) have all their voices been heard and full datasets (transcriptions) been saturated; there is no indication as to whether the excerpts presented derived from pre-school or primary school (nor subject area) - might this be important? Is your sample more representative of the former e.g., 12 versus 8 ... findings could therefore be presented much more transparently and more meaningfully contextualised
It is clear to see how some discussion points and conclusions drawn can be traced back to the evidence (narrative excerpts) presented, yet by no means all. There is a great need to ensure that all claims being made can be substantiated by the evidence
Very careful editorial work required throughout to pick up lapses in sentence structure, spelling and the use of appropriate expressions ... this would enable the reader to follow the line of argument and points being raised far more easily. Ensure that all citations can be traced in references e.g., Kvale (2011) appears to be missing
Author Response
Point 1: The term 'reflexive' needs to be changed in the title to 'reflective' - these two words are not interchangeable and do not mean one and the same thing
Response 1: The correction is made
Point 2: Review keywords beneath the abstract
Response 2: The keywords are reviewed and some more relevant to the content of the article are incorporated
Point 3. Strengthen theorisation underpinning the use of such concepts as reflection, reflective practice, professional knowledge, pedagogical knowledge and key tenets of tutoring.
Response 3. A section with theoretical references is incorporated considering the suggestion made.
Point 4. Strengthen methodology section - hermeneutic phenomenology requires greater clarity and depth; what does the literature say about in-depth interviews; what are the ethical issues related to the use of zoom as your platform and how were these mitigated; how was data stored and confidentiality, anonymity assured; what questions were the participants asked .
Response 4. The details of the observation made are taken into account and more information is incorporated into the methodology.
Point 5. Section 3 - how were the narrative excerpts selected (beyond the use of computer software); which pedagogical programme and university did your interview participants attend ... with approximately 50 minutes (x20 participants) have all their voices been heard and full datasets (transcriptions) been saturated; there is no indication as to whether the excerpts presented derived from pre-school or primary school (nor subject area) - might this be important? Is your sample more representative of the former e.g., 12 versus 8 ... findings could therefore be presented much more transparently and more meaningfully contextualised
Response 5. In Table 1 of the article (p. 6) you can see the details about the pedagogical program and the university. The name of each university is not provided based on ethical criteria. All the voices of the 20 participants were heard, both in the direct application of the interview and after the transcriptions were made. In turn, throughout the analysis process, the recordings were repeatedly reviewed and the stories provided by the participants were listened to. An acronym is added at the end of each interview to identify which program it corresponds to. Regarding your observation on the proportion of the sample (12 versus 8), a comparison was not made in this regard because it was not considered within the objectives of the study. In the research problem, more antecedents of the context are incorporated to favor the understanding of the reality studied and an attempt is made to present the results in a more organized way. At all times we tried to comply with the criteria of transparency.
Point 6. It is clear to see how some discussion points and conclusions drawn can be traced back to the evidence (narrative excerpts) presented, yet by no means all. There is a great need to ensure that all claims being made can be substantiated by the evidence
Response 6. Adjustments are made to the discussion and conclusions in an attempt to ensure that the claims back up to the results.
Point 7. Very careful editorial work required throughout to pick up lapses in sentence structure, spelling and the use of appropriate expressions ... this would enable the reader to follow the line of argument and points being raised far more easily. Ensure that all citations can be traced in references e.g., Kvale (2011) appears to be missing
Response 7. The original article is sent to a new translation. The bibliographical references are reviewed, taking care that all those that were cited in the body of the work are present.
Reviewer 5 Report
Some sound findings although the presentation of these needs to be improved and consideration needs to be given to wider context of reflection in terms of written submissions and the role of the questions in the findings.
Other comments made on attached document.

Author Response
Point 1: Some sound findings although the presentation of these needs to be improved and consideration needs to be given to wider context of reflection in terms of written submissions and the role of the questions in the findings.
Response 1. The results are presented in a more orderly manner and the questions asked in the interview are incorporated into the methodology (see Table 2, p. 6).
Point 2. Other comments made on attached document.
Response 2. They carefully review each of the observations and take them into consideration for the restructuring of the article
Round 2
Reviewer 1 Report
The Author has labored to re-work on the manuscript as evidenced by the substantial amount of new text. I want to state that the new text is better than what was presented in the first draft.
That said, I still want to stress that the manuscript still suffers from language issues. There are several instances where the author uses words that mean something else other than what the author wishes/intends to communicate. I have made a number of comments throughout the entire draft, though not exhaustively (see attached).
The author claimed as part of his/her response to the earlier draft that she/he had received language editing support. I want to suggest that the language be checked again, especially regarding to the proper use of words and terms in sentences. As an example, it is difficult to understand what the author implies/mean, when she/he writes ".... to the configuration of pedagogical knowledge....." the use of the term "configuration" is problematic when one is talking about individuals and how they acquire or develop knowledge. There are several examples of this nature in the manuscript.
There are a number of studies /articles where teacher knowledge is addressed, especially the so-called pedagogical content knowledge. Moreover, there are studies that have characterized teacher knowledge into different forms, distinguishing teacher knowledge elements acquired during teacher training and teacher knowledge elements attained through and with practice during practicum for example. I am surprised that the author has not taken up or considered these recent and more credible literature, when she/he discussing what constitutes pedagogical knowledge.
In summary, the language issue should be addressed to the level that is acceptable to the target audience. Moreover, the sentences should not be so long as they are in most cases.

Author Response
Point 1: The Author has labored to re-work on the manuscript as evidenced by the substantial amount of new text. I want to state that the new text is better than what was presented in the first draft.
Response 1: Thank you very much for your comment
Ponit 2. : That said, I still want to stress that the manuscript still suffers from language issues. There are several instances where the author uses words that mean something else other than what the author wishes/intends to communicate. I have made a number of comments throughout the entire draft, though not exhaustively (see attached).
Response 2: Words that were not understandable were changed, also trying to simplify the wording of some ideas that from their perspective were not clear.
I appreciate the time spent on his detailed review of the language and all the contributions made. I hope that now the ideas I want to communicate are clearer.
Ponit 3: The author claimed as part of his/her response to the earlier draft that she/he had received language editing support. I want to suggest that the language be checked again, especially regarding to the proper use of words and terms in sentences. As an example, it is difficult to understand what the author implies/mean, when she/he writes ".... to the configuration of pedagogical knowledge....." the use of the term "configuration" is problematic when one is talking about individuals and how they acquire or develop knowledge. There are several examples of this nature in the manuscript.
Response 3: I did indeed receive support for the translation of the manuscript, both from a professional in the field of education who is fluent in English and from the editing service offered by the journal. However, I believe that beyond the fact that there may be certain details in the translation, there are certain words that acquire certain meanings when used in different contexts. In any case, I made changes in an attempt to find synonyms that were more universal. Throughout the document, the word "configuration" was changed.
Ponit 4: There are a number of studies /articles where teacher knowledge is addressed, especially the so-called pedagogical content knowledge. Moreover, there are studies that have characterized teacher knowledge into different forms, distinguishing teacher knowledge elements acquired during teacher training and teacher knowledge elements attained through and with practice during practicum for example. I am surprised that the author has not taken up or considered these recent and more credible literature, when she/he discussing what constitutes pedagogical knowledge.
Response 4: Indeed, there is currently a significant body of literature that addresses the category of pedagogical content knowledge, proposed by Lee Shulman, which is understood to be a category of pedagogical content knowledge., therefore, it is a type of knowledge that articulates disciplinary and pedagogical knowledge. However, the perspective considered in the work that I present is based on the notion of "knowledge", understanding this concept from the way in which teachers make use of theoretical, experiential, personal and contextual knowledge to carry out their pedagogical work.
In any case, I considered your suggestion and incorporated some internationally cited authors, but also various Latin American and Chilean authors who for some time have been making important contributions to teacher training, taking into account the socio-political and educational reality of the territory.
I am very grateful for the suggestion of the sources he recommended, I only considered one of them, the most current one.
Point 5: In summary, the language issue should be addressed to the level that is acceptable to the target audience. Moreover, the sentences should not be so long as they are in most cases.
Response 5: Suggested adjustments were made. Thank you for your comments
Reviewer 2 Report
I am happy that the authors have made significant improvements. But still, care needs to be taken to the details including but not limited to the following items: different parentheses were used for citation ((), [], (]), the numbering of subsection titles in the results section is not correct, and text/paragraph alignment needs to be revised (e.g., last sentence right before the table 2)
Author Response
Point 1: I am happy that the authors have made significant improvements. But still, care needs to be taken to the details including but not limited to the following items: different parentheses were used for citation ((), [], (]), the numbering of subsection titles in the results section is not correct, and text/paragraph alignment needs to be revised (e.g., last sentence right before the table 2)
Response 1: Thank you very much for your comments, they were taken into account to optimise the manuscript.
Reviewer 4 Report
Whilst comments made on the first iteration of this manuscript have, for the most part, been given due consideration the following points really do need to be addressed before this second iteration can be published:
- theorisation of core concepts being drawn upon - the work of key scholars (and seminal texts) who have contributed much to the field are not referenced at all and although research questions have been identified these are not grounded within a robust theoretical framework - this in turn, has major implications for the frames of reference used during the analysis and synthesis of data collected
- minor lapses in sentence structure still evident so would benefit from further editorial work
Anomalies detected in the revised version submitted:
- Figure 1 – ‘Tutor teacher’ used in version 1 whereas in version 2 this is presented as ‘university tutor teacher’
- Discrepancy found in many of the narrative excerpts presented which raises the question of credibility and accurately re-presenting participant voice verbatim, such as:
Version 1
All time I was being evaluated and the truth is that it is something that I really appreciate because when someone is teaching there are many things that one does not realize, that may not be correct or that I have not explained well or even used the right words, so, the role of the teacher has been very important (27:12 ¶ 45 in Interview 9).
Version 2
When you are teaching, there are many things you don’t actually realise, that may not be correct or that you might not have explained properly, or maybe I didn't use the most accurate words, so yes, her role has been critical (27:12 ¶ 45 in Interview 9/ECE).
Version 1
You have to submit a reflection report on the practice, you have to give some information about the school and the grade you are teaching and to include, for instance, our own appreciation of how we feel with the class that was taking classes of english (15:55 ¶ 105 in Interview 2).
Version 2
You must submit a reflection report on the practice, you must give some details of the establishment and specific data of the course in which we developed our practice and include, for example, our appreciation of how we felt with the course that took the English classes (15:55 ¶ 105 in Interview 2/PE).
Author Response
Whilst comments made on the first iteration of this manuscript have, for the most part, been given due consideration the following points really do need to be addressed before this second iteration can be published:
Point 1. theorisation of core concepts being drawn upon - the work of key scholars (and seminal texts) who have contributed much to the field are not referenced at all and although research questions have been identified these are not grounded within a robust theoretical framework - this in turn, has major implications for the frames of reference used during the analysis and synthesis of data collected
Response 1: The theoretical references underpinning the research presented in the article were studied in depth and some adjustments were made in response to this request. Thank you very much for your comments
Point 2. minor lapses in sentence structure still evident so would benefit from further editorial work
Response 2. Adjustments were again made to the wording of the original document and to the English translation.
Point 3. Anomalies detected in the revised version submitted:
- Figure 1 – ‘Tutor teacher’ used in version 1 whereas in version 2 this is presented as ‘university tutor teacher’
Response 3. The name of the category was adjusted in the software (Atlas.ti), so that it would be understood in the image that it is the university tutor and not the school tutor, as in some countries, for example Spain, the school teacher is called a tutor. That was the only purpose, if in your opinion it is necessary to leave the initial image, I have no problem.
Point 4. Discrepancy found in many of the narrative excerpts presented which raises the question of credibility and accurately re-presenting participant voice verbatim, such as:
Version 1
All time I was being evaluated and the truth is that it is something that I really appreciate because when someone is teaching there are many things that one does not realize, that may not be correct or that I have not explained well or even used the right words, so, the role of the teacher has been very important (27:12 ¶ 45 in Interview 9).
Version 2
When you are teaching, there are many things you don’t actually realise, that may not be correct or that you might not have explained properly, or maybe I didn't use the most accurate words, so yes, her role has been critical (27:12 ¶ 45 in Interview 9/ECE).
Response 4. You are absolutely right and I am grateful to you for pointing that out. The problem arose because those who helped me with the translation of the document made adjustments to the wording, I think with the purpose of making it more understandable to the reading public, which I did not realise when I uploaded the file for the second revision, so I take responsibility for that. In no case was the intention to alter the faithful account of the participants, I fully understand the relevance of the criteria of rigour of a scientific research to be valid, in addition to the safeguard of the ethical criteria that must be present.
Fully sharing your observation and thanking you once again, the corresponding adjustments were made to the quotations, respecting the literacy of the original speech given by the participants.
Reviewer 5 Report
There is a great improvement shown in this manuscript which the author(s) should be commended for. Some small changes are suggested for the final version:
* abstract - line 8 - condensation is not the right word for this sentence
* please go through the manuscript and check sentence length. At times, there are extensive sentences - sometimes the whole paragraph. This impacts on readability to check where you use 'because' or a ; and see if this can be a full stop and then a new sentence to break up these large blocks of writing.
* check the heading level for 3.1. Value Attributed...
* Where is 3.2? You have 3.1, 3.1.1, 3.1.2 then 3.3. Should this be 3.2?
Best wishes for the publication.
Author Response
Point 1. There is a great improvement shown in this manuscript which the author(s) should be commended for. Some small changes are suggested for the final version:
Response 1. Thank you very much for your appreciation
Point 2. abstract - line 8 - condensation is not the right word for this sentence
Response 2. The concept of phenomenologically based meaning condensation was extracted from the proposal of Steinar Kvale (201) in his book "Las entrevistas en la Investigacion Cualitativa" (Interviews in Qualitative Research). Madrid, Spain. Morata, pp. 140-142. However, for the sake of comprehensibility in the abstract , the concept of "meaning condensation" was changed to "categories of meaning".
Point 3. please go through the manuscript and check sentence length. At times, there are extensive sentences - sometimes the whole paragraph. This impacts on readability to check where you use 'because' or a ; and see if this can be a full stop and then a new sentence to break up these large blocks of writing.
Response 3. Adjustments were made to the manuscript taking into consideration the observation made
Point 4. * check the heading level for 3.1. Value Attributed...* Where is 3.2? You have 3.1, 3.1.1, 3.1.2 then 3.3. Should this be 3.2?
Response 4. The requested adjustments were made
Best wishes for the publication.
Thank you very much for your good wishes